



# Delayed and rapid deglaciation of alpine valleys in the Sawatch Range, southern Rocky Mountains, USA

Joseph P. Tulenko[1], William Caffee[1], Avriel D. Schweinsberg[1], Jason P. Briner[1], Eric M.

Leonard[2].

1. Department of Geology, University at Buffalo, Buffalo, NY 14260, USA

2. Department of Geology, Colorado College, Colorado Springs, CO 80903, USA

## Abstract

We quantify retreat rates for three alpine glaciers in the Sawatch Range of the southern Rocky Mountains following the Last Glacial Maximum using [10]Be ages from ice-sculpted, valley-floor bedrock transects and statistical analysis via the BACON program in R. Glacier retreat in the Sawatch Range from at (100%) or near (~83%) Last Glacial Maximum extents initiated between 16.3 and 15.6 ka and was complete by 14.2 – 13.7 ka at rates ranging between 9.9 and 19.8 m a$^{-1}$. Deglaciation in the Sawatch Range commenced ~2 – 3 kyr later than the onset of rising global $CO_2$, but approximately in-step with rising temperatures observed in the North Atlantic region at the Heinrich Stadial 1/Bølling transition. Our results highlight a possible teleconnection between the North Atlantic sector and the southern Rocky Mountains. However, deglaciation in the Sawatch Range also approximately aligns with the timing of Great Basin pluvial lake lowering. Recent data-modeling comparison efforts highlight the influence of the large North American ice sheets on climate in the western United



States, and we hypothesize that recession of the North American ice sheets may have

influenced the timing and rate of deglaciation in the Sawatch Range. While we cannot

definitively argue for exclusively North Atlantic forcing or North American ice sheet

forcing, our data demonstrate the importance of regional forcing mechanisms on past

climate records.

## 1. Introduction

Alpine glaciers worldwide underwent substantial retreat in response to climate

warming during the last deglaciation (Shakun et al., 2015; Palacios et al., 2020).

However, the general trend of warming through the last deglaciation was interrupted by

internally forced and regionally heterogeneous climate changes such as the cool

Heinrich Stadial 1 (17.5 – 14.7 ka), abrupt warming into the Bølling-Allerød period (14.7

– 12.9 ka), and the Younger Dryas cold period (12.9 – 11.7 ka) all centered in the North

Atlantic region (NGRIP members, 2004; Rasmussen et al., 2014). To thoroughly

characterize the influence of these climatic oscillations, their expression throughout the

Northern Hemisphere is often investigated using records of mountain glaciation (Ivy-

Ochs et al., 2006; Schaefer et al., 2006; Young et al., 2011; Shakun et al., 2015;

Marcott et al., 2019; Young et al., 2019). Mountain glacier deposits serve as suitable

archives since mountain glaciers are particularly sensitive to changes in climate (e.g.

Oerlemans, 2005; Roe et al., 2017). Furthermore, where deposits are carefully mapped

and dated, quantitative retreat or thinning rates of glaciers can be compared to records

of climatic forcings. Using statistical approaches to quantify retreat and thinning rates

has been previously applied to ice sheets (e.g., Johnson et al., 2014; Koester et al.,



2017; Lesnek et al., 2020) but only for a few mountain glaciers (e.g., Hofmann et al, 2019).

In the western United States (US; Fig. 1), mountain glaciers expanded out of the high elevations of the Rocky Mountains, the Sierra Nevada, the Uinta Mountains, and many other, smaller ranges during the Last Glacial Maximum (LGM; Porter et al., 1983; Pierce, 2003). During the last deglaciation, many glaciers retreated from their extended LGM positions and eventually melted from their cirques by the start of the Holocene (e.g., Marcott et al., 2019). Yet, the temporal and spatial patterns of retreat throughout the western US and their relationship to hemispheric and global forcing are still a subject of debate. Glaciers in the western US may have retreated in response to rising global atmospheric $CO_2$ concentrations, thus broadly synchronous with other mountain glaciers around the world (e.g. Shakun et al., 2015; Marcott et al., 2019). However, some evidence suggests a delay of deglaciation until the Bølling due to either persistent stadial conditions (e.g., Young et al., 2011) or as a response to increased local moisture supply to some glaciers from nearby pluvial lakes (e.g. Laabs et al., 2009).

Over a decade of work has resulted in detailed moraine chronologies in three adjacent alpine valleys in the Sawatch Range of central Colorado (Fig. 2; Briner, 2009; Young et al., 2011; Shroba et al., 2014; Leonard et al., 2017b; Schweinsberg et al., 2020). While these studies primarily focused on mapping and dating the range-front moraines and associated outwash terraces, a transect of ages from bedrock samples in Lake Creek valley (Fig. 2) documented rapid retreat between 15.6 ± 0.7 ka and 13.7 ± 0.2 ka (Leonard et al., 2017b; Schweinsberg et al., 2020). Schweinsberg et al. (2020) suggested a possible link between North Atlantic climate forcing and the rapid



deglaciation observed in Lake Creek valley, but similar transects from adjacent valleys

are lacking to bolster or refute this hypothesis.

**Figure 1**. Key moraine chronologies from the southern Rocky Mountains and locations of glaciation centers and large pluvial lakes in the western US following the
Last Glacial Maximum (LGM). LL = Lake Lahontan, LB = Lake Bonneville, A = Colorado Front Range, B = Sangre de Cristo Mountains, C = San Juan Mountains, and D = Winsor Creek valley, New Mexico. The largest star corresponds to our field site in the Sawatch Range. LGM ice limits from Dalton et al. (2020). Inset is of the western portion of North America. CIS = Cordilleran Ice Sheet, LIS = Laurentide Ice Sheet.



Here, we combine 12 new cosmogenic [10]Be exposure ages with ten previously

published [10]Be ages from bedrock samples along transects in three adjacent alpine

valleys in the Sawatch Range, southern Rocky Mountains (Fig. 2). By dating bedrock

sites along valley transects, we characterize the timing and pace of glacier retreat

during the last deglaciation. We calculate rates of deglaciation for each valley with best-

fit time-distance plotting using the R program BACON (Fig. 4). Our results suggest that

glaciers in the Sawatch Range may have been influenced more heavily by regional

forcing than by global $CO_2$ concentrations.

## 2. Setting

The high peaks of south-central Colorado and northern New Mexico compose

the southern end of the Rocky Mountain Range in North America and were home to

many alpine glaciers during multiple glaciations throughout the Pleistocene (Fig. 1;

Pierce, 2003; Leonard et al., 2017b; Marcott et al., 2019). Transects of [10]Be ages from

bedrock along valley axes exist for a few valleys in the upper Boulder Creek drainage in

the Front Range, Colorado (Benson et al., 2004; Ward et al., 2009; Dühnforth and

Anderson, 2011). While some evidence from the Boulder Creek drainages may suggest

delayed deglaciation, chronologic scatter in the ages makes it difficult to determine the

exact timing and how quickly glaciers retreated to their cirques. Existing ages from one

valley the Sangre de Cristo Range, south-central Colorado, suggest that a glacier there

remained near its LGM terminus until ~16 ka, but then retreated to its cirque in a period

of ~2 kyr (Leonard et al., 2017a). In the Animas River valley of the San Juan Mountains,

southwest Colorado, existing [10]Be ages indicate glacier retreat began as early as ~19



ka, with retreat of half of the total valley length occurring from ~16 – 13.5 ka (Guido et al., 2007). Relatively early initial retreat of the glacier in the Animas River valley is

contingent on dating at a single site. Near Baldy Peak in Northern New Mexico, LGM moraines and what appear to be cirque moraines have been surveyed in the Winsor Creek valley (Armour et al., 2002; Marcott et al., 2019). [10]Be ages from the cirque, ~4 km up-valley from the LGM moraines, range from 15.8 – 14.3 ka, suggesting that the glacier retreated to near its cirque within that interval. The recessional and LGM

moraines remain undated so it is difficult to know when the glacier began retreating. In summary, while there is some chronologic scatter in ages from these sites, there is evidence to suggest that some glaciers in the southern Rocky Mountains remained relatively expanded through the beginning of the last deglaciation and were delayed in their retreat. However, once retreat was underway, all sites observed thus far reveal

that glaciers completely retreated at least up to their cirques prior to the Younger Dryas cold period with no evidence for subsequent moraine deposition.

Prominent moraines originally mapped as part of the surficial geologic map of the Granite 7.5' quadrangle (updated by Shroba et al., 2014) exist at the mouths of multiple glacially sculpted valleys within the Sawatch Range. Of these, moraines deposited at

the mouths of three adjacent valleys, Lake Creek, Clear Creek and Pine Creek, have been thoroughly surveyed and dated (Fig. 2; Briner, 2009; Young et al., 2011; Schweinsberg et al., 2020). The moraine chronologies reveal that following the LGM, which culminated between ~22 and 19 ka, glaciers remained at (100%) or near (82 – 83%) their LGM lengths until ~16 – 15 ka, after which the moraine record stops; in all





three valleys, no moraines have yet been found farther up-valley. Young et al. (2011)

argued

**Figure 2**. Ice-sculpted bedrock [10]Be ages from Lake Creek (LC; orange), Clear Creek
(CC; green), and Pine Creek valleys (PC; blue). Included are LGM and recessional
moraines (solid colored lines) with reported ages for the LGM moraine in Pine Creek
valley at 16.3 ± 0.4 ka (n=x; Young et al., 2011) and a recessional moraine in Lake
Creek valley at 15.6 ± 0.7 ka (n=x; Schweinsberg et al., 2020). There is a similar,



undated recessional moraine in Clear Creek valley that we hypothesize is also ~16 ka.
Ice-sculpted bedrock ages reported here include analytical uncertainty, and moraine
ages are reported as mean and one standard deviation. Glaciers at their mapped LGM
extents are delineated in gray.

that since all three glaciers are east-facing and in close proximity—yet show differences

in the timing of LGM culmination between the valleys—it is possible that non-climatic

factors, such as glacier hypsometry, may have influenced the timing and extent of LGM

culminations. While there are pre-existing [10]Be ages measured in a transect along Lake

Creek Valley that track the retreat of the glacier through the last deglaciation, the other

two valleys have not yet been surveyed. As such, it remains unclear if glacier

hypsometry also influenced the timing and pace of deglaciation between all three

adjacent valleys.

### 3. Methods and materials

Sample collection for [10]Be dating from Clear Creek and Pine Creek valleys was

conducted in the summers of 2017 and 2018. Twelve samples were collected from

exposed, glacially sculpted bedrock surfaces along the Clear Creek (n=8) and Pine

Creek (n=4) valley floors, spanning from just within range-front moraines up to each

respective cirque (Figs. 2 and 3). Samples were processed at the University at Buffalo

Cosmogenic Isotope Laboratory following slightly modified versions of quartz

purification and beryllium extraction procedures refined at the University of Vermont

(Corbett et al., 2016). After quartz purification, samples were dissolved in acid along

with a [9]Be carrier spike. Beryllium was then purified and extracted, oxidized, and

packed into targets for measurement at the Center for Accelerated Mass Spectrometry

at Lawrence Livermore National Laboratory. [10]Be/[9]Be ratios were measured and





standardized to the reported 07KNSTD3110 ratio of 2.85 x $10^{-12}$ (Nishiizumi et al.,

2007). Our 12 ages and 10 previously published ages were calculated using the Cronus

Earth online calculator (developmental version 3;

https://hess.ess.washington.edu/math/index_dev.html; Balco et al., 2008). We calculate

ages using the Promontory Point production rate (Lifton et al., 2015) and the LSD$n$

scaling model (Lifton et al., 2014) – a combination used extensively throughout the

western US (e.g., Licciardi and Pierce, 2018; Quirk et al., 2018; Brugger et al., 2019;

Schweinsberg et al., 2020). Below, we discuss in more detail how different production

rate calibrations and scaling schemes impact our results.

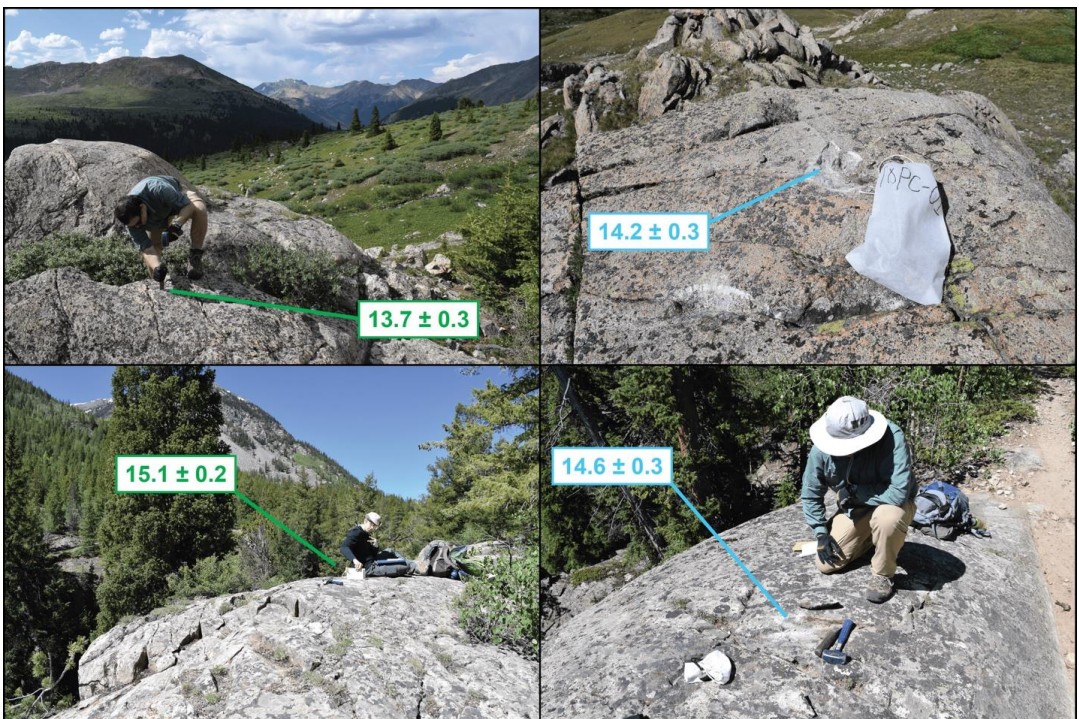

**Figure 3**. Field photos of ice-sculpted bedrock surfaces from selected locations.
Clockwise from top left: 18CC-04, 18PC-01, 18PC-05, 17CC-08. Color scheme for ages
matches Figures 2 and 4: Clear Creek valley samples = green; Pine Creek valley
samples = blue.



To calculate retreat rates, we used the BACON program in R (Blaauw and

Christen, 2011). This program generates age-depth models for stratigraphic records

based on chronologic constraints at various depths. The model takes each age and the

depth of the constraint as inputs. The model then interpolates between each point using

Bayesian analysis and the geologic principle of superposition to build an age-depth

model of sedimentation rate with a statistical treatment of uncertainty. Here, we use the

[10]Be ages measured in each valley and their geographic coordinates as positions along

the length of each respective valley floor, where the toe of the glacier at the LGM is the

starting point (e.g., 100% or maximum length), and the base of each valley's cirque wall

is the end point (e.g., 0% or minimum length). The retreat rates presented here are net

retreat rates, although it is possible there may have been short-lived re-advances that

did not lead to significant moraine deposition. BACON outputs a time series of age-

length points and 95% confidence intervals.

### 4. Results

All 22 sculpted-bedrock [10]Be ages, which span from immediately inboard of the

innermost moraine to the cirque floors, range between 16.0 ± 0.4 and 13.5 ± 0.3 ka (Fig.

2, Table 1). In Lake Creek valley, seven ages span from 67.1 – 1.2% of the distance of

the valley floor, ranging between 15.2 ± 0.4 and 13.5 ± 0.3 ka. Nine [10]Be ages spanning

from 67.7 – 0.8% in Clear Creek valley range between 15.3 ± 0.2 and 13.7 ± 0.2 ka. In

Pine Creek valley, six [10]Be ages span from 78.2 – 2.4% and range between 16.0 ± 0.4

and 14.2 ± 0.3 ka. In addition to the bedrock ages from each valley, a recessional

moraine at 82% of the LGM position sampled in the Lake Creek system dates to 15.6 ±


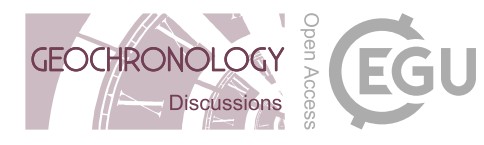

Table 1. Sample data and ${}^{10}$Be ages.

| Sample Name | Latitude (DD) | Longitude (DD) | Elevation (m asl) | Thickness (cm) | Shielding correction | ${}^{10}$Be concentration (atoms/g) | ${}^{10}$Be concentration err. (atoms/g) | ${}^{10}$Be age (ka)[a] | ${}^{10}$Be age (ka)[b] | Transect dist. (m) | Transect dist. (%) |
|---|---|---|---|---|---|---|---|---|---|---|---|
| Lake Creek transect | | | | | | | | | | | |
| AR09-01 | 39.06590 | -106.40703 | 2930 | 1.5 | 0.9960 | 513652 | 12462 | 15.2 ± 0.4 | 16.5 ± 0.4 | 22430 | 67.1 |
| AR09-10 | 39.07059 | -106.47182 | 3048 | 3.5 | 0.9890 | 508740 | 13867 | 14.4 ± 0.4 | 15.6 ± 0.4 | 16531 | 49.5 |
| AR09-11 | 39.10098 | -106.54449 | 3261 | 4.0 | 0.9830 | 543494 | 10771 | 13.5 ± 0.3 | 15.0 ± 0.3 | 7278 | 21.8 |
| LKCK-15-3 | 39.15210 | -106.52745 | 3761 | 3.0 | 0.9830 | 767000 | 10200 | 14.0 ± 0.2 | 15.8 ± 0.2 | 950 | 2.8 |
| LKCK-15-1 | 39.15800 | -106.53100 | 3740 | 1.5 | 0.9859 | 760192 | 9960 | 13.8 ± 0.2 | 15.6 ± 0.2 | 500 | 1.5 |
| LKCK-15-2 | 39.15809 | -106.53175 | 3776 | 4.0 | 0.9848 | 807000 | 11600 | 14.7 ± 0.2 | 16.5 ± 0.2 | 500 | 1.5 |
| LKCK-15-4 | 39.14907 | -106.52420 | 3774 | 3.0 | 0.9740 | 750000 | 12000 | 13.7 ± 0.2 | 15.5 ± 0.2 | 412 | 1.2 |
| Clear Creek transect | | | | | | | | | | | |
| AR09-03 | 39.00260 | -106.33924 | 2835 | 2.0 | 0.9920 | 462709 | 9308 | 14.8 ± 0.3 | 15.9 ± 0.3 | 18756 | 67.7 |
| 17CC-06 | 39.00270 | -106.35680 | 2928 | 2.0 | 0.9821 | 508593 | 7026 | 15.3 ± 0.2 | 16.7 ± 0.2 | 17058 | 61.6 |
| 17CC-08 | 38.99920 | -106.36590 | 2916 | 1.0 | 0.9668 | 490889 | 5908 | 15.1 ± 0.2 | 16.4 ± 0.2 | 16118 | 58.2 |
| 17CC-11 | 38.99720 | -106.37550 | 2954 | 1.5 | 0.9643 | 494089 | 6071 | 14.9 ± 0.2 | 16.2 ± 0.2 | 15251 | 55.1 |
| 17CC-09 | 38.98970 | -106.41250 | 3049 | 1.5 | 0.9815 | 523681 | 9768 | 14.7 ± 0.3 | 16.0 ± 0.3 | 11527 | 41.6 |
| 17CC-10 | 38.98970 | -106.42410 | 3096 | 2.0 | 0.9780 | 527682 | 6338 | 14.5 ± 0.2 | 15.8 ± 0.2 | 10459 | 37.8 |
| 18CC-01 | 38.94677 | -106.45814 | 3317 | 2.0 | 0.9645 | 583017 | 10873 | 14.1 ± 0.3 | 15.6 ± 0.3 | 3455 | 12.5 |
| 18CC-02 | 38.93217 | -106.45944 | 3403 | 2.5 | 0.9863 | 631265 | 11746 | 14.3 ± 0.3 | 15.8 ± 0.3 | 1776 | 6.4 |
| 18CC-04 | 38.91955 | -106.46294 | 3625 | 2.5 | 0.9851 | 691699 | 12979 | 13.7 ± 0.3 | 15.3 ± 0.3 | 208 | 0.8 |
| Pine Creek transect | | | | | | | | | | | |
| AR09-07 | 38.97437 | -106.25060 | 2931 | 3.0 | 0.9960 | 536779 | 14280 | 16.0 ± 0.4 | 17.5 ± 0.5 | 14345 | 78.2 |
| AR09-08 | 38.97437 | -106.25060 | 2931 | 1.0 | 0.9960 | 537781 | 10676 | 15.8 ± 0.3 | 17.2 ± 0.3 | 14345 | 78.2 |
| 18PC-06 | 38.96590 | -106.27515 | 3244 | 1.5 | 0.9816 | 643484 | 14156 | 15.8 ± 0.3 | 17.4 ± 0.4 | 11377 | 62.0 |
| 18PC-05 | 38.94764 | -106.33513 | 3403 | 3.0 | 0.9661 | 631502 | 11710 | 14.6 ± 0.3 | 16.2 ± 0.3 | 5528 | 30.1 |
| 18PC-01 | 38.91623 | -106.36673 | 3824 | 1.5 | 0.9899 | 818218 | 15138 | 14.2 ± 0.3 | 16.1 ± 0.3 | 519 | 2.8 |
| 18PC-02 | 38.91648 | -106.36528 | 3827 | 2.5 | 0.9899 | 880955 | 18751 | 15.3 ± 0.3 | 17.4 ± 0.4 | 438 | 2.4 |

Notes: Rock density for all samples 2.65 g/cm³; zero surface erosion rate applied to all samples

[a]Ages calculated using the Promontory Point production rate calibration (Lifton et al., 2015) and LSDn scaling (Lifton et al., 2014)

[b]Ages calculated using the Northeast North America production rate clibration (Balco et al., 2009) and Lm scaling (Lal, 1991; Stone, 2000)



0.7 ka (Schweinsberg et al., 2020). There is a similar-appearing moraine at 83% of the LGM position in Clear Creek valley. Although it is undated, we correlate this moraine in Clear Creek valley to the moraine dated to 15.6 ± 0.7 ka in Lake Creek valley. There is

no recessional moraine in Pine Creek valley, but a cluster of ages at 16.3 ± 0.4 ka from the LGM moraine suggest that the glacier re-advanced to or remained at its LGM extent until nearly the same time when glaciers in the other two valleys deposited recessional moraines (Briner, 2009; Young et al., 2011).

         Most ages in each valley are in stratigraphic order and fall within the 95%
confidence interval calculated in BACON, except for four ages (Fig. 4). Ages from Lake Creek valley suggest the glacier retreated from its recessional moraine position (82%) at 15.6 ± 0.7 ka, and reached its cirque (~1.2%) by 13.7 ± 0.2 ka. Clear Creek valley ages suggest the glacier retreated from its recessional moraine position (83%) at 15.6 ± 0.7 ka and reached its cirque (0.8%) by 13.7 ± 0.3 ka. Finally, Pine Creek valley ages

suggest the glacier was at its LGM extent (100%) until 16.3 ± 0.4 ka and then retreated to its cirque (2.8%) by 14.2 ± 0.2 ka.

         Results from BACON analysis suggest the net retreat rate for the glacier in Lake Creek valley between 15.6 ± 0.7 ka (Schweinsberg et al., 2020) and 13.7 ± 0.2 ka averages 19.8 ± 10.0 m a$^{-1}$ (Fig. 4). The net retreat rate calculated from BACON for the

glacier in Clear Creek valley between 15.6 ± 0.7 ka and 13.7 ± 0.3 ka averages 11.1 ± 3.7 m a$^{-1}$. Finally, the net retreat rate for the glacier in Pine Creek valley from the LGM position at 16.3 ± 0.4 ka (Young et al., 2011) to 14.2 ± 0.3 averages 9.9 ± 5.7 m a$^{-1}$.

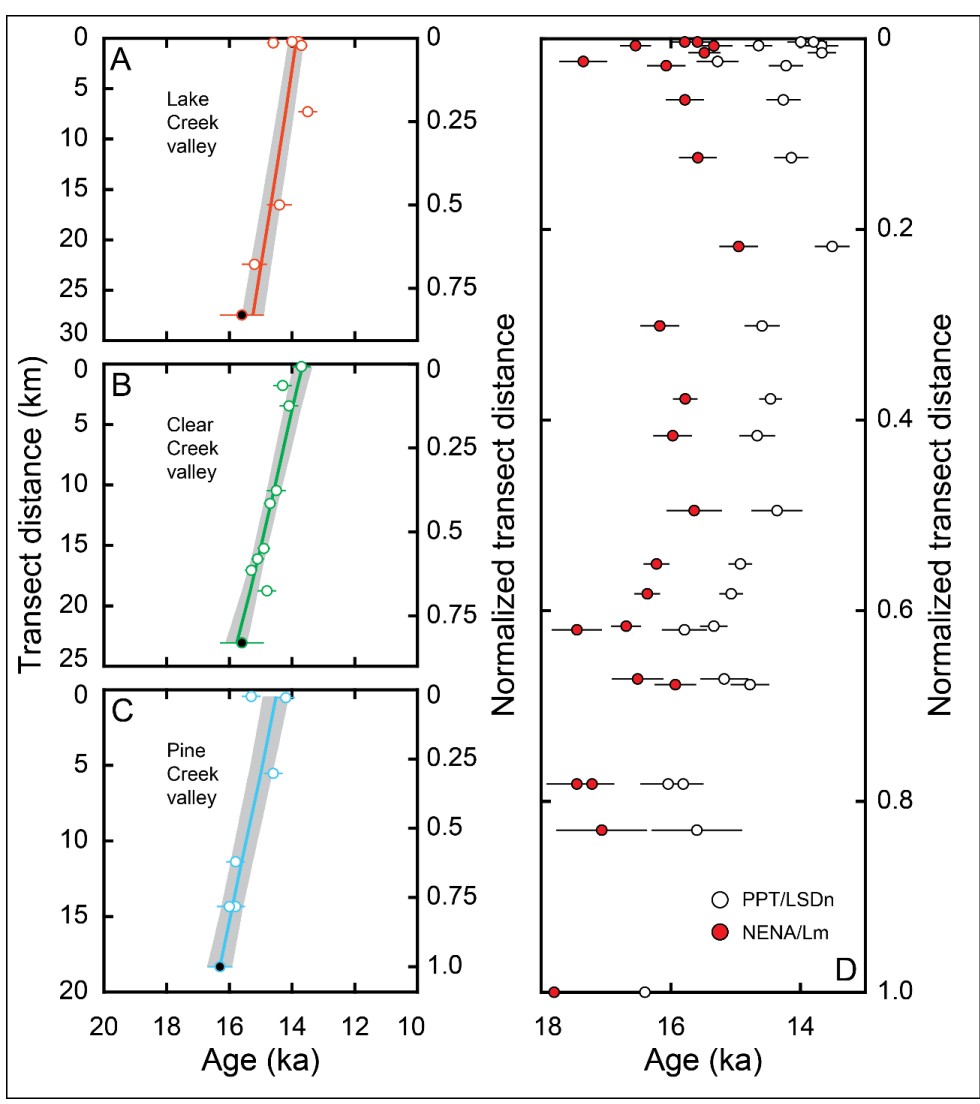

**Figure 4**. Summary plots of [10]Be ages and BACON statistical analysis results. A) Lake
Creek valley (orange), B) Clear Creek valley (green), and C) Pine Creek valley (blue).
Ages in solid black fill at the bottom of each transect are from recessional moraine ages
(Young et al., 2011; Schweinsberg et al., 2020). BACON results are mean (color lines)
and 95% confidence intervals (gray shading). Left y-axes are total valley floor distances
from the LGM moraine to the base of each respective cirque headwall (note that scales
are different because valley lengths are different). Right y-axes are the same, but
normalized values, where 1 = LGM moraine position and 0 = base of cirque headwall.
D) Distribution of all ages using both PPT (Lifton et al., 2015) and LSD*n* (Lifton et al.,



2014), and NENA (Balco et al., 2009) and Lm (Lal, 1991; Stone, 2000) production rate calibration and scaling scheme combinations discussed in the text.


## 5. Discussion

### 5.1 Reliability of bedrock ages

While most bedrock ages along each valley transect are in stratigraphic order, we find four ages that do not comply with stratigraphic order and fall outside the 95%

confidence interval of the retreat rates calculated from BACON. For example, in Lake Creek valley, one age at 13.5 ± 0.3 ka is younger than all up-valley ages, which average 13.8 ± 0.2 ka (excluding one possible outlier outside of the BACON 95% confidence interval). In Clear Creek valley, the age from the farthest downvalley site of 14.8 ± 0.3 ka may be a possible outlier because the next three ages up-valley are all older and in

stratigraphic order, the oldest of which is 15.3 ± 0.2 ka. Finally, one age from the Pine Creek cirque of 15.3 ± 0.3 ka may be an outlier because it is older than the next age downvalley (14.6 ± 0.3 ka) as well as a second sample from the cirque of 14.2 ± 0.3 ka.

Although we interpret our results using the Promontory Point production rate calibration site (Lifton et al., 2015) and the LSDn scaling scheme (Lifton et al. 2014), we

calculate exposure ages using another commonly used calibration site that is from northeastern North America (NENA; Balco et al., 2009) and another commonly used scaling scheme (Lal/Stone–Lm; Lal, 1991; Stone, 2000). Samples used for the NENA production rate calibration range in elevation between ~50 to 400 m asl and are located ~3000 km northeast of the Sawatch Range. This combination produces ages between 9

to 12% older (Fig. 4; Table 1). We do not feel confident in calculating exposure ages using other production rate calibration sites since the sites in closest proximity likely





shared the most similar exposure histories. Ultimately, we favor the Promontory Point

production rate calibration site (Lifton et al., 2015) because the site is closest in both

location (site is ~600 km from the Sawatch Range) and elevation (sample elevations are

~1600 m asl) to our study area.

*5.2 The last deglaciation of the Sawatch Range and the southern Rocky*

*Mountains*

The pattern of deglaciation in both Clear Creek valley and Pine Creek valley

appears to follow the pattern previously observed in Lake Creek valley (Young et al.,

2011; Leonard et al., 2017b; Schweinsberg et al., 2020). All three glaciers remained at

(100%) or near (82 – 83%) their LGM extents until 16.3 – 15.6 ka, after which all three

glaciers rapidly retreated to their cirques within the next ~2 kyr, at rates ranging

between 19.8 and 9.9 m a$^{-1}$. It is possible that the glacier in Pine Creek valley began

retreating ~500 yr earlier than the other two glaciers, and likewise completely

deglaciated ~500 yr earlier. Pine Creek valley is shorter and steeper than the other two

valleys. Thus, it is possible that variations in valley hypsometry between Pine Creek

valley and the other two valleys may have caused the slight difference in their

deglaciation histories. We conclude that while there may have been some hypsometric

influences on the timing of deglaciation across our study site, evidence suggests these

influences were minimal. We find that all three valley glaciers did not begin significantly

retreating until ~5 – 6 kyr after the culmination of the LGM in the Sawatch Range;

however, once glacier retreat initiated, deglaciation was completed within ~2 kyr.



From the existing records in the southern Rocky Mountains synthesized above, we find that the pattern of deglaciation observed in the Sawatch Range was consistent in a few but not all sites across the region. Collecting more records of alpine deglaciation in the southern Rocky Mountains may be necessary to further test which pattern, if any, is the dominant pattern of deglaciation in the region.

*5.3 Drivers of southern Rocky Mountain deglaciation*

Records of global climate change over the last deglaciation suggest a link between rising $CO_2$ concentrations and global temperature (Denton et al., 2010; Shakun et al., 2012; Putnam et al., 2013). However, there is noticeable spatial heterogeneity in both the timing and magnitude of warming through the last deglaciation that cannot be attributed to global $CO_2$ forcing alone (e.g., Clark et al., 2012). We find that the initiation of significant deglaciation in some locations across the southern Rocky Mountains lagged rising $CO_2$ concentrations by as much as ~2 – 3 kyr (Fig. 5), which suggests these glaciers were more likely influenced by regional forcings rather than global $CO_2$.

Ice core records—among other records—reveal a complex pattern of abrupt warming and cooling events that occurred in the North Atlantic region during the last deglaciation (Fig. 5; Buizert et al., 2014). Despite rising $CO_2$ concentrations beginning ~18 ka, North Atlantic records reveal that cold conditions persisted until 14.7 ka, known as Heinrich Stadial 1 (HS-1). Following these sustained cold conditions, an abrupt transition to warmer conditions is marked by the HS-1/Bølling boundary at 14.7 ka (Buizert et al., 2014). We find that the timing of abrupt warming documented in the North Atlantic at the HS-1/Bølling transition aligns somewhat closely with the timing of





**Figure 5**. Deglaciation of the Sawatch Range compared to other climate proxies. From top to bottom: Atmospheric $CO_2$ concentrations (Bereiter et al., 2015); Synthesized Greenland temperature from ice cores (Buizert et al., 2014); Lake Level reconstructions of Lake Bonneville (LB; black dashed line) and Lake Lahontan (LL; gray dashed line) from Reheis et al. (2014); Normalized BACON plots from Lake Creek (LC; orange), Clear Creek (CC; green) and Pine Creek valleys (PC; blue). Vertical lines correspond to the onset of $CO_2$ rise beginning ~18 ka and the Heinrich Stadial 1/Bølling transition at 14.7 ka.

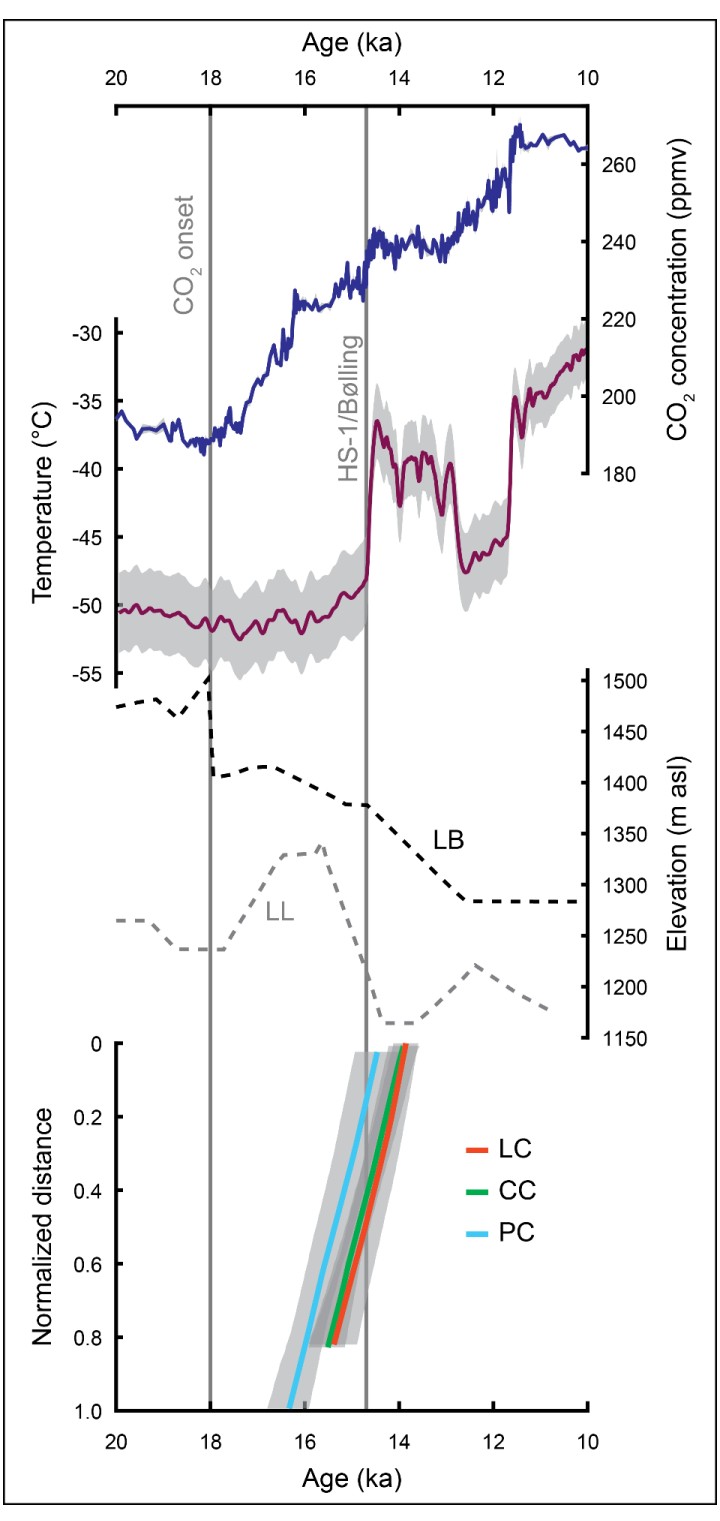

deglaciation in the

southern Rocky

Mountains. Additionally,

we find that the rapid rate

of deglaciation following a

period of relative glacier

stability concurs with the

drastic North Atlantic shift



from cold stadial conditions to significant warming. The similarity between alpine glacier records in the southern Rocky Mountains and North Atlantic climate history indicates a possible teleconnection between the two regions.

In addition to the alpine glaciers that existed in the mountainous regions of the western US during the late Pleistocene, large pluvial lakes such as Lake Lahontan and Lake Bonneville existed across the Great Basin (Fig. 1; Gilbert, 1890; Russell, 1885; Orme, 2008). These lakes could have been sustained by increased precipitation delivery to the southwestern US (e.g., Munroe and Laabs, 2013; Oster et al., 2015; Lora

and Ibarra, 2019) or were maintained simply by colder temperatures persisting throughout the region (e.g., Benson et al., 2013). Recent syntheses of past Great Basin lake levels reveal that Lahontan and Bonneville lakes resided at relative high stands between 15.5 and 14.5 ka (Benson et al., 2013; Reheis et al., 2014; Oviatt, 2015). After this time, each lake experienced notable declines in lake level (Fig. 5), which could have

been the result of reduced precipitation due to re-arranging storm tracks, warming temperature or a combination of both (Benson et al., 2013; Oster et al., 2015; Lora and Ibarra, 2019).

Recent modeling efforts have highlighted how North American ice sheets likely influenced atmospheric circulation and regional climate throughout the Pleistocene

(COHMAP members, 1985; Lofverstrom et al., 2014; Liakka and Lofverstrom, 2018; Lora and Ibarra, 2019). Specifically, there appears to have been drastic shift in climatologies over the western US when the Cordilleran (CIS) and Laurentide (LIS) ice sheets separated (Lofverstrom et al., 2014; Lora et al., 2016; Tulenko et al., 2020). For example, during the last deglaciation, once the CIS and LIS separated, some model



results suggest the western US became warmer and drier (Lora et al., 2016). The latest

synthesis of the last deglaciation of the major North American ice sheets suggests the

separation occurred between 16 and 15 ka (Dalton et al., 2020). Thus, it is possible that

the saddle collapse and separation of the CIS and LIS and resulting atmospheric re-

organization may have led to both drastic pluvial lake level reductions and the rapid

deglaciation of some glaciers in the southern Rocky Mountains.

Between North Atlantic forcing and North American ice sheet forcing, it is difficult

to conclude what the primary driver of deglaciation in the Sawatch Range was; it may

be a combination of both forcings. We find that the approximate timing and rate of

deglaciation observed in the Sawatch Range points to abrupt warming and/or drying,

and is supported by pluvial lake level records in the western US, which have also been

tied to both North Atlantic forcing and North American ice sheet forcing (Munroe and

Laabs, 2013; Benson et al., 2013; Lora and Ibarra, 2019). Regardless, the data

synthesized here underscore the dominance of regional forcing mechanisms over global

forcing mechanisms on some climate records in the western US.


### 6. Conclusions

We constrain the timing and rate of deglaciation in three alpine valleys in the

Sawatch Range, southern Rocky Mountains. Beryllium-10 ages from ice-sculpted

bedrock in each valley reveal the significant retreat of glaciers from their LGM extents

(100%) or near (82 – 83%) their LGM extents was initiated shortly after 16.3 – 15.6 ka,

despite ~2 – 3 kyr of prior global warming forced by rising atmospheric $CO_2$. Glaciers in

three adjacent valleys retreated rapidly to their cirques within ~2 kyr, culminating at





~14.2 – 13.7 ka, at rates ranging between 19.8 to 9.9 m a$^{-1}$. We recognize that using the

NENA production rate and Lm scaling produces ages 9 – 12% older than the ages

reported herein, which would change the interpretation of the dataset. However, we

favor the PPT/LSD*n* combination because the PPT calibration site is closest in proximity

and elevation to the Sawatch Range.

        We hypothesize that one of two possible regional mechanisms were responsible

for driving the pattern of deglaciation for some glaciers in the southern Rocky

Mountains. First, we find that some alpine glaciers in the region began retreating around

the time of abrupt warming observed at the Heinrich-Stadial 1/Bølling transition. In

addition to the timing, the relatively rapid and short-lived nature of retreat for some

glaciers – including those in the Sawatch Range – across the southern Rocky

Mountains is more consistent with the abrupt manner of warming observed in the North

Atlantic than with global $CO_2$ forcing. Alternatively, lake level records reveal that both

Bonneville and Lahontan lakes lowered nearly in step with some retreating alpine

glaciers across the southern Rocky Mountains. Previous studies have linked Great

Basin pluvial lake regression to warming and the migration of prevailing storm tracks

due to atmospheric re-organization that may have been forced by separation of North

American ice sheets. Thus, warming and drying induced by abrupt atmospheric re-

organization at the time of LIS and CIS separation may have driven both Great Basin

lake level lowering and rapid alpine glacier retreat in some valleys in the southern

Rocky Mountains. While we cannot conclude that either one of the aforementioned

forcing mechanisms was solely responsible for deglaciation of the Sawatch Range, we

suggest that either one or both were stronger controls than global $CO_2$ forcing.



**Acknowledgements**

JP Tulenko acknowledges funding from University at Buffalo Geology
Department Duane Champion Fund for field work. AD Schweinsberg acknowledges
funding from the Colorado Scientific Society and Mark Diamond Research Fund from
the University at Buffalo Graduate Student Association for field work. We also thank AJ
Lesnek for assistance in the 2018 field campaign, and CM Russell and RK Kroner for
sample collection in the Lake Creek valley cirque in 2015.

**Author contributions**

**JP Tulenko:** Investigation (sample collection and processing), Conceptualization, Data
curation, Writing – original draft, Visualization; **W Caffee:** Investigation (sample
collection and processing), Writing – review and editing; **AD Schweinsberg:**
Investigation (sample collection and processing), Conceptualization, Writing – review
and editing; **JP Briner:** Investigation (sample collection and processing),
Conceptualization, Data curation, Supervision, Funding acquisition; **EM Leonard:**
Investigation (sample collection), Conceptualization, Data curation, Writing – review and
editing.

**Competing Interests**

The authors declare that they have no known competing financial interests or personal
relationships that could have appeared to influence the work reported in this paper.

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
