# Peer review of "Delayed and rapid deglaciation of alpine valleys in the Sawatch Range, southern Rocky Mountains, USA"

_Geochronology, 2020_

## Short Comment (SC1) · 13 May 2020

Thank you for this nice contribution on glacier dynamics. In the introduction you state that "mountain glacier deposits serve as suitable archives since mountain glaciers are particularly sensitive to changes in climate". As we all know, this is mostly due to their short response time to climatic variations. One major problem is, however, that factors other than climate, such as topography, may lead to glacier advances or stationary periods of (small) mountain glaciers during a general trend of climatic amelioration. Examples are widespread in the literature, see Lukas et al., the Holocene (https://doi.org/10.1177/0959683607078983) for an example. In the discussion you

mention that the valley hypsometry may have led to a different pace of deglaciation in the valleys. You argue, however, that this influence should be regarded as negligible. Could you elaborate this a little bit further and provide some arguments for excluding a significant topographic forcing?

---

## Referee Comment (RC1) · Richard Selwyn Jones (Referee) · 8 Jun 2020

**Summary and recommendation**

The paper reports a combination of new and previously published geochronological data (10Be ages) in the southern Rocky Mountains, which record glacier retreat following the last glacial maximum. The authors use the data to quantify retreat rates, and then discuss possible drivers of glacier retreat.

Overall, I enjoyed reading the paper, which adds new data and analysis in this region, and which presents interesting questions about possible global, regional and

local drivers of alpine deglaciation. The manuscript is generally well structured, written and illustrated. The methods are all outlined, although more detail could be provided in parts. The results broadly support the conclusions, but more discussion and clarity are required.

I recommend publication in Geochronology following some revision.

There are a few areas where I think the paper could be improved. First, a key component of this work is the quantification of retreat rates, but the authors could better emphasise why knowing the glacier retreat rates is important. Arguably, the discussion about the timing and drivers of deglaciation are supported by the 10Be ages without the need for rates. Second, glacier hypsometry is the only the non-climatic factor that is considered, and this is only briefly discussed. Other non-climatic drivers should be acknowledged and discussed. Third, the comparison to the North Atlantic (Heinrich-Stadial 1/Bolling) climate transition requires either more justification or less weight as a conclusion. The chronology indicates that the glaciers retreated prior to the abrupt warming of the transition, implying that this didn't, at least initially, drive glacier retreat. In summary, this is a nice dataset that, while not conclusive about what drove deglaciation in this region or the extent of possible teleconnections, presents an opportunity to thoroughly discuss possible drivers of glacier retreat.

**Detailed comments**

Lines 123-125: It is not clear from this sentence whether the estimate that "glaciers remained at (100%) or near (82-83%) their LGM length until 16-15 ka" is derived by this study or a previous study.

Lines 139-145: Hypsometry is the only named non-climatic factor that is considered. Were these glaciers ever lake-terminating (e.g. Lake Creek)? What role could bed geometry have played?

Line 150: The text refers to a "slightly modified" method. Modified from what – Corbett

et al. (2016)? Modified how?

Line 155-160: What was the ratio/10Be concentration of your procedural lab blank(s)?

Lines 175-179: This isn't the first time that Bayesian age-depth models have been used for transects of 10Be ages. Previous such work (e.g. Jones et al., 2015, Nat. Comms.; Small et al., 2018, GSA Bull.) should be acknowledged. In general, the approach to derive retreat rate estimates needs more detail. What is exactly being modelled here? Is it assuming a linear or non-linear relationship between age and depth/distance? Is the model accounting for age uncertainties? If so, are the age uncertainties included at 1 or 2 standard deviations, weighted or unweighted?

Lines 183-184: Clarify what you mean by "net retreat rates".

Results: You should initially report the results for only the new data (the 12 ages from Clear Creek and Pine Creek), even if only described briefly. After that, you can describe the results in combination with the previously published ages.

Line 190 (and elsewhere): How confident are you in the precision of your distance measurements? Would rounding to the nearest whole percent be more suitable?

Line 214 (and elsewhere): Please clarify here whether the retreat rate result is reported at 68% or 95% confidence. Additionally, the format of reporting is probably not suitable, as the model output distribution is likely non-Gaussian. Such results are therefore typically reported as an uncertainty range, rather than mean with uncertainty.

Lines 233-243: The identification of likely outliers is based on the general stratigraphic relationship of ages within the dataset. These outliers also happen to fall outside of the 95% confidence bounds from the BACON model. But, as far as I can tell, BACON was not used to systemically identify (and remove) outliers. In which case, the estimated retreat rates from BACON will be influenced by these apparent outliers. So, how do the retreat rates differ when these outliers are excluded?

Lines 247-252: More of a discussion point than a criticism: While it seems fairly well

justified to use the Promontory Point calibration site instead of NENA site based on locality and elevation range, it is also worth considering the time period used for the calibration sites. The Promontory Point site is calibrating the production rate at 18.9-18.0 ka, while the NENA site is calibrating for 16.8-13.8 ka. The dataset reported here best correlates to the time period covered by the NENA site, which could be an argument to use this production rate instead of that from Promontory Point.

Lines 266-269: Explain how glacier hypsometry and/or steepness would influence differing glacier behaviour during deglaciation.

Lines 285-288: Glaciers don't respond to CO2, so directly comparing to CO2 seems a little irrelevant. Of course, there is a close relationship between CO2 and temperature, but why not compare your glacier retreat records to proxy global temperature (e.g. Shakun et al., 2012)?

Lines 327-334: The argument that there is similarity between records, and "possible teleconnections", isn't particularly convincing. The majority of the recorded retreat occurred before the North Atlantic climate shift; your ages indicate retreat initiated 1-2 kyr earlier that the climate shift at 14.7 ka. I'd like to see the text rephrased, without mention of teleconnections.

Lines 329-330: What is this period of relative glacier stability based on?

Line 383: "one of two", or both mechanisms, as you state below. Reword this, as these are not mutually exclusive explanations.

Lines 385-386: As mentioned above, it is a difficult to accept that the glaciers "began retreating around the time of abrupt warming" when the data indicate retreat started at least 1-2 kyr before the climate transition. There is only correlation here if you doubt the accuracy (or precision) of the retreat ages, in which case you should discuss more thoroughly.

Lines 387-390: I like the comparison of the rates of glacier and climate change, as it

makes use of your estimated retreat rates and it can be effective if there is any doubt in the absolute timing. However, a number of non-climatic, glaciological processes can also contribute to faster rates, even with a gradual forcing. Such processes also need to be considered.

Line 400: Sorry to be pedantic, but as above, glaciers don't respond to gas concentrations. Refer to global temperature instead.

Table 1: Transact distances are reported to the nearest metre over many thousands of metres. This seems unrealistically precise.

Figure 2: Need to make clear what are the new data and what are previously published data. There are also two references to "n=x", which I presume need values added.

---

## Referee Comment (RC2) · Benjamin Laabs (Referee) · 10 Jun 2020

Summary: The authors report a new set of cosmogenic 10Be exposure ages along the retreat path in well-studied glacial valleys in the Sawatch Range in southern Colorado. The data are especially interesting because they provide limits on the rate of ice retreat at the end of the last Pleistocene glaciation and show remarkable similarity. The authors have done an excellent job of interpreting the ages in the context of existing cosmogenic chronologies of glacial deposits elsewhere in the region and assess the regional vs. global climate forcings that likely affected the retreat of glacier termini during the last glacial-interglacial transition.

[Figure]

I believe the manuscript is suitable for publication in Geochronology and will be of broad interest to the readership. I suggest the following revisions, especially concerning the need for (1) additional explanation of the potential limitations and sources of error in exposure ages of glacially scoured bedrock, and (2) the comparison of exposure ages computed with the NENA calibration and the Promontory Point calibration.

Line-by-line comments:

Line 50: the Uinta Mountains are part of the Middle Rocky Mountains physiographic province and probably do not need to be singled out here (although they are awesome and have a fantastic glacial record).

Line 53: could probably state "Latest Pleistocene or Early Holocene" here, as Marcott et al. found that some cirque floor moraines were abandoned as early as 15 ka. Additionally, basal 14C ages from lake sediments inboard of cirque-floor moraines are Pleistocene in age in some mountains (see records published by J. Munroe for the Uinta Mountains (Munroe and Laabs, 2017) and by J. Munroe and others in the Ruby Mountains).

Lines 90-93: should cite some of the earlier, original reports on the glacial record in southern Colorado and northern New Mexico. Jim McCalpin did some work in the region (mostly the Sangres) in the 1980s and Keith Brugger has done more recent mapping in the Sawatch.

Lines 101-104: the Guido et al. cosmo ages are pre-CRONUS (and also pre-really good AMS measurements) and probably should be recalculated in order to accurately compare with more recently published cosmo ages from southern Colorado. If you've already done this, then it's worth specifying here. If not, the Guido et al. ages are available in ICE-D.

Line 174: prior to this paragraph, consider adding a paragraph about how exposure ages of glacially scoured bedrock are related to ice margin position and some potential limitations of dating these to track ice retreat compared to moraines. As you know, glacially scoured bedrock surfaces that protrude above the valley floor (forming smooth and easy-to-sample surfaces) represent places of minimal scour depth, which can result in an inheritance problem. The Bayesian approach helps to sort this out by accounting for relative age differences, but even so, the potential for inheritance is greater than for most other applications of cosmogenic dating and should be acknowledged. Snow cover is another important consideration along valley floors and should be acknowledged if not assessed.

Lines 189-211: consider reorganizing the reporting of ages here. The bedrock exposure ages are reported first, then the exposure ages of recessional moraines/young modes of terminal moraines, and then the bedrock ages are described again. Perhaps starting with the moraine ages (or including them in a previous section) and focusing just on the bedrock exposure ages here would improve the flow of this section and a smoother transition to the retreat rates in the subsequent paragraph.

Lines 233-242: the statistical reasons for excluding four exposure ages are explained well here, but the most likely reason that some exposure ages fall out of stratigraphic order, inconsistent exposure between sample sites, is not. As noted in a previous comment, the potential limitations of bedrock exposure ages should be acknowledged.

Lines 243-255: I can't see the reason for using NENA-Lm as an example of another production/scaling model for high altitude sites in western NA. The NENA calibration site is far away and much lower in elevation, and I think the reason for using it in some earlier studies in the mountain west was to illustrate the effects of lower SLHL production rate (which started to appear in the literature circa 2010) on exposure ages. Perhaps a better option would be to compare the ages computed with the Promontory Point calibration/LSDn scaling with ages computed with a globally averaged production rate and LSDn scaling, or just show the effects of using different scaling models with the Promontory Point calibration? This would better illustrate the degree to which the choice of production rate affects exposure ages, which I assume is what the authors

are doing here.

Lines 267-269: should probably cite Young et al. (2011) at the end of this sentence.

Fig. 1: this is a beautiful map! As you reference some other glaciated mountains in the western U.S. in the introductory paragraphs, consider labeling some of the ones shown on the map along with pluvial lakes.

Fig. 2: seems like a good idea to show all the terminal moraine cosmo ages instead of just the young mode at Pine Creek, given that the terminals are the "starting point" for ice retreat? Just a suggestion; I understand that you're emphasizing the onset of ice recession in this paper, not the glacier maxima.

Fig. 5: may want to consider a more recent and focused assessment of the Bonneville hydrograph in Oviatt (2015) or some of the specific discussions about the duration of the Provo phase of the lake by D. Miller (2016).

---

## Author Comment (AC1) · 15 Jun 2020

Reply to SC1 'forcing of glacier dynamics' from Felix Martin Hofmann

We thank the author for their comment regarding valley hypsometry, and the implication that the small differences in deglaciation reconstructions (timing, rate) between our valleys were partly attributed to this and or other non-climatic forcings. While we did not specifically use the term 'negligible', we do recognize that a 500 yr offset between Pine Creek valley and the other two valleys is notable, and each valley has slightly different net retreat rates. Our original phrasing could better reflect that:

[Figure]

Line 269 "We conclude that while there may have been some hypsometric influences on the timing of deglaciation across our study site, evidence suggests these influences were minimal."

In the example cited, Lukas et al. argue that topographic shielding led to a delay/standstill in the deglacial pattern for that particular glacier (i.e. not forced by climate). Here, it is curious that deglaciation initiated first for the glacier (Pine Creek valley glacier) that we might expect to have been slightly better-shielded by topography compared to the other two paleo-glaciers, which occupied larger and broader valleys. This appears contradictory to the argument that non climatic topographic shielding played a significant role. Regardless, we find that although the Pine Creek paleo-glacier may have initiated its pulse of recession ~500 yr sooner than the other two, all three paleo-glaciers experienced a period of ~1-1.5-kyr-long synchronous retreat once the other two glaciers began retreating.

In addition, we observe that the rates of retreat in all three valleys differ. We wonder if the rates are different as a result of valley hypsometry: the paleo-glacier in Pine Creek valley – which has the steepest average valley gradient at 65 m/km – retreated the slowest, and the paleo-glacier in Lake Creek valley retreated the fastest (37 m/km). This makes sense because a shallower and broader glacier is more sensitive to changes in ELA. We suggest this line of evidence demonstrates that valley hypsometry impacted the pace of retreat in a predictable way, which is worth highlighting.

Combined, we feel that there is sufficiently strong enough evidence to support the conclusion that while there were likely some non-climatic factors that influenced the timing of initiation and rate of retreat for these glaciers, climatic forcing is largely responsible for the significant, ~1-1.5 kyr synchronous retreat event that took place in all three valleys.

We are adding in a brief discussion of retreat rates and average valley gradients and how the two scale, as well as appending the concluding statement highlighted at the

beginning of our response:

Starting at Line 217: "The calculated average valley gradients for each valley – measured as the elevation change divided by the horizontal length of each valley bottom transect from LGM moraine up to the base of each respective cirque – are 29 m/km for Lake Creek valley, 37 m/km for Clear Creek valley, and 65 m/km for Pine Creek valley."

Starting at Line 269: "We also observe that Pine Creek valley has the steepest average valley gradient and the slowest net retreat rate, which is predictably a direct result of valley hypsometry since glacier lengths in steeper valleys generally adjust less to equivalent changes in ELA. On the other hand, glaciers occupying the lower-gradient Lake and Clear creek valleys experienced higher reconstructed rates of retreat. Regardless, we find that while Pine Creek may have initiated $\sim$500 yr sooner than the other two, all three valleys were in a period of $\sim$1-1.5-kyr-long synchronous retreat once the other two glaciers began retreating. We conclude that while there may have been some hypsometric influences on the timing of deglaciation across our study site, evidence suggests these influences did not keep these glaciers from synchronously retreating during a majority of their deglaciation."

---

## Editor Comment (EC1) · Yeong Bae Seong (Editor) · 16 Jun 2020

Dear Authors,

Most of all, thanks for submitting a well-written paper. The dataset should be very interesting and robust enough to be shared with other geo-scientists working in the area and similar settings.

I read through the comments by a guest and two reviewers, most of which are relevant and productive. As you can see, the two reviewers are very positive at the manuscript and the dataset but do not like the some parts of the present content. I hope you can

make good responses and modification, point by point.

I would add below some personal comments as well, which are not overlap with the ones by reviewers.

- Ln 271-2: "We find that all three valley glaciers did not begin significantly retreating until ∼5 − 6 kyr after the culmination of the LGM in the Sawatch Range". You cannot jump over the gun like above because you are based on the different types of samples (i.e. Moraine erratics VS bedrock). Moraine boulders can indicate advance or stagnation (As you know there is some debate on the implication of ages of boulders on a moraine. Polished bedrock usually indicates the timing of deglaciation as you did. There should be differentiation on the interpretation of ages of two types sampled (Reviewer 2 told about this problem). You may want to make more explanation and discussion on this matter.

Figure 2: Can you separate the type of samples for 10Be dating? For example, erratics (open circle) Vs bedrock (filled circle).

Figure 3: How about showing the sample number on the picture (or on the sampled boulder or bedrock), which is better to readers?

I hope the authors would like to incorporate most of comments raised, which should be conducive to improve the present manuscript.

Best, Associate Editor Yeong Bae Seong.

---

## Author Comment (AC2) · 22 Jul 2020

Reply to 'Review' from RC 1 Richard Selwyn Jones

We thank RC 1 for their thoughtful and constructive review of our manuscript. We have gone through and replied to each individual comment. Below, please find the original comments bolded and italicized and our reply in normal font.

***Detailed comments***

***Lines 123-125: It is not clear from this sentence whether the estimate that "glaciers remained at (100%) or near (82-83%) their LGM length until 16-15 ka" is derived by this study or a previous study.***

While we measured the normalized moraine locations in this paper, the age ranges for the moraines between 16-15 ka were established in previous studies that dated those moraines. We have re-arranged this portion of the text, in accordance with both yours and RC 2's suggestion to read:

"The moraine chronologies reported thus far reveal that following the LGM (which culminated between ~22 – 19 ka), a recessional moraine at 82% of the LGM position sampled in the Lake Creek system was deposited at 15.6 ± 0.7 ka (Schweinsberg et al., 2020). There is a similar-appearing moraine at 83% of the LGM position in Clear Creek valley. Although it is undated, we tentatively correlate this moraine in Clear Creek valley to the moraine dated to 15.6 ± 0.7 ka in Lake Creek valley. Finally, there is no recessional moraine in Pine Creek valley, but a cluster of ages at 16.0 ± 0.9 ka from the LGM moraine suggest that the glacier re-advanced to or remained at its LGM extent until nearly the same time when glaciers in the other two valleys deposited recessional moraines (Briner, 2009; Young et al., 2011)."

***Lines 139-145: Hypsometry is the only named non-climatic factor that is considered. Were these glaciers ever lake-terminating (e.g. Lake Creek)? What role could bed geometry have played?***

While we recognize that there are lakes present in both Lake Creek and Clear Creek, these are dammed reservoirs and may or may not have been present at the time of deglaciation. Whether or not glaciers may have been lake-terminating near their terminal moraines, our focus is on glacier retreat upvalley, which would not have been influenced by terminal-moraine-area lake effects. Thus, the majority of retreat commenced in these valleys without any possible influence from lake-terminating dynamics. Even if lakes did exist, and dynamics associated with lakes is likely not among the explanations for any inter-valley variability.

To the point of bed geometry, we should be more explicit in describing how the retreat rates are influenced by valley geometries/hypsometry. We find that the retreat rates are significantly different between each valley and we wonder if this is attributable to valley geometry/hypsometry. The smallest, shortest and steepest valley (Pine Creek) retreats at the slowest rate while the other two valleys which are broader and much larger

retreat at faster rates. As mentioned in our response to SC 1, we find there may have been some notable non-climatic factors that influenced when glaciers began initially retreating, and perhaps even the rates of retreat among the different valleys, but we think that the ~1 – 1.5 kyr of synchronous retreat across all three valleys is strong evidence for a climatic driver.

As mentioned in our reply to the short comment in the open discussion, we are adding in a brief discussion of retreat rates and average valley gradients and how the two scale, as well as appending the concluding statement at Line 269:

Starting at Line 217: "The calculated average valley gradients for each valley – measured as the elevation change divided by the horizontal length of each valley bottom transect from LGM moraine up to the base of each respective cirque – are 29 m/km for Lake Creek valley, 37 m/km for Clear Creek valley, and 65 m/km for Pine Creek valley."

Starting at Line 269: "We also observe that Pine Creek valley has the steepest average valley gradient and the slowest net retreat rate, which is predictably a direct result of valley hypsometry since glacier lengths in steeper valleys generally adjust less to equivalent changes in ELA. On the other hand, glaciers occupying the lower-gradient Lake and Clear creek valleys experienced higher reconstructed rates of retreat. Regardless, we find that while Pine Creek may have initiated ~500 yr sooner than the other two, all three valleys were in a period of ~1-1.5-kyr-long synchronous retreat once the other two glaciers began retreating. We conclude that while there may have been some hypsometric influences on the timing of deglaciation across our study site, evidence suggests these influences did not keep these glaciers from synchronously retreating during a majority of their deglaciation."

***Line 150: The text refers to a "slightly modified" method. Modified from what – Corbett et al. (2016)? Modified how?***

The differences in the procedure between our lab and the UVM lab are very minor and likely not worth mentioning. We have elected instead to simply remove the phrase "slightly modified"

***Line 155-160: What was the ratio/$^{10}$Be concentration of your procedural lab blank(s)?***

We updated the text to include that information:

"After quartz purification, samples were dissolved in acid along with a $^{9}$Be carrier spike in two batches each with a process blank."

And

"For samples collected in 2018, the process blank $^{10}Be/^{9}Be$ ratio was 2.96 x 10-15, and for samples collected in 2017 the process blank $^{10}Be/^{9}Be$ ratio was 9.56 x 10-16 (see Table 1 for details on sample collection dates)."

In addition, we added a footnote to Table 1 listing the process blank values.

***Lines 175-179: This isn't the first time that Bayesian age-depth models have been used for transects of $^{10}Be$ ages. Previous such work (e.g. Jones et al., 2015, Nat. Comms.; Small et al., 2018, GSA Bull.) should be acknowledged. In general, the approach to derive retreat rate estimates needs more detail. What is exactly being modelled here? Is it assuming a linear or non-linear relationship between age and depth/distance? Is the model accounting for age uncertainties? If so, are the age uncertainties included at 1 or 2 standard deviations, weighted or unweighted?***

Thank you for bringing these additional citations to our attention. We tried to be as thorough in acknowledging that this type of work has been published before so we are happy to include these citations in the list cited at the end of the first introduction paragraph.

As to describing BACON with slightly more detail, we amended the mentioned paragraph to now read:

"To calculate retreat rates, we used the BACON program in R (Blaauw and Christen, 2011). This program generates age-depth models for stratigraphic records based on chronologic constraints at various depths. Here, we use the 10Be ages and their 1-sigma uncertainties measured in each valley as the age input and the geographic coordinates of each age as the depth inputs. The position along the valley floor is scaled such that the toe of the glacier at the LGM is the starting point (e.g., 100% or maximum length), and the base of each valley's cirque wall is the end point (e.g., 0% or minimum length). The model then interpolates between each point using Bayesian analysis and the geologic principle of superposition to build an age-length model with an unweighted statistical treatment of uncertainty. The interpolation between points is smoothed (i.e. non-linear) based on retreat rates at previous positions. The retreat rates presented here are net retreat rates, because it is possible there may have been short-lived re-advances that did not lead to significant moraine deposition. BACON outputs a time series of age-length points and non-Gaussian 95% confidence intervals. Calculated retreat rates are assumed to be linear, and we report the 95% uncertainty range."

***Lines 183-184: Clarify what you mean by "net retreat rates".***

We use the term "net" retreat rates because there may be many short-term re-advances "hidden" in our chronology, short events that are un-detectable by our chronology. Thus actual retreat rates could have been higher locally. We believe that "net" retreat rate is an appropriate way to characterize our derived retreat rates.

***Results: You should initially report the results for only the new data (the 12 ages from Clear Creek and Pine Creek), even if only described briefly. After that, you can describe the results in combination with the previously published ages.***

We added the following sentence to the beginning of the paragraph:

"The 12 new sculpted-bedrock 10Be ages reported here range 15.8 ± 0.3 – 13.7 ± 0.3 ka (Fig. 2; Table 1)."

We also update Table 1 and Figure 2 to identify which samples are new and which samples are previously published.

***Line 190 (and elsewhere): How confident are you in the precision of your distance measurements? Would rounding to the nearest whole percent be more suitable?***

We measured profiles along valley floors in ArcGIS to get precise numbers, but we agree that rounding to the nearest whole percent is more realistic. We fixed this throughout the text and in Table 1.

***Line 214 (and elsewhere): Please clarify here whether the retreat rate result is reported at 68% or 95% confidence. Additionally, the format of reporting is probably not suitable, as the model output distribution is likely non-Gaussian. Such results are therefore typically reported as an uncertainty range, rather than mean with uncertainty.***

We appreciate the insight on reporting model outputs that you correctly pointed out are non-Gaussian. We now report the 95% uncertainty range throughout the text.

***Lines 233-243: The identification of likely outliers is based on the general stratigraphic relationship of ages within the dataset. These outliers also happen to fall outside of the 95% confidence bounds from the BACON model. But, as far as I can tell, BACON was not used to systemically identify (and remove) outliers. In which case, the estimated retreat rates from BACON will be influenced by these apparent outliers. So, how do the retreat rates differ when these outliers are excluded?***

When calculating the resulting retreat rates if we remove outliers and in all three valleys, the retreat rates decrease by 1.7, 2.7 and 6% for Lake Creek, Clear Creek and Pine Creek valleys respectively. We have added the following sentence:

"Removal of potential outliers reduces retreat rates by 1.7%, 2.7% and 6% for Lake Creek, Clear Creek, and Pine Creek valleys respectively."

***Lines 247-252: More of a discussion point than a criticism: While it seems fairly well justified to use the Promontory Point calibration site instead of NENA site based on locality and elevation range, it is also worth considering the time period***

*used for the calibration sites. The Promontory Point site is calibrating the production rate at 18.9- 18.0 ka, while the NENA site is calibrating for 16.8-13.8 ka. The dataset reported here best correlates to the time period covered by the NENA site, which could be an argument to use this production rate instead of that from Promontory Point.*

This is a good point – production rate choice is always a topic of discussion. Fundamentally, this is why we provide our ages with two reasonable production rate choices. As the reviewer knows, ultimately, you have to choose one to go with for the main text. While we agree that there are advantages to dating features close in age to a calibration site, it is likely that other factors (as mentioned in the text) like site elevation, are also important. Ultimately, the age ranges at the PPT and NENA calibration sites are fairly close in age to ours. That said, because PPT is the closest in elevation to our field area, and is a rate that others are using in their papers for Rocky Mountain cosmogenic nuclide chronologies, we chose to report PPT in our text.

**Lines 266-269: Explain how glacier hypsometry and/or steepness would influence differing glacier behaviour during deglaciation.**

We are adding in a brief discussion of retreat rates and average valley gradients and how the two scale, as well as appending the concluding statement highlighted at the beginning of our response:

Starting at Line 217: "The calculated average valley gradients for each valley – measured as the elevation change divided by the horizontal length of each valley bottom transect from LGM moraine up to the base of each respective cirque – are 29 m/km for Lake Creek valley, 37 m/km for Clear Creek valley, and 65 m/km for Pine Creek valley."

Starting at Line 269: "We also observe that Pine Creek valley has the steepest average valley gradient and generally the slowest net retreat rate, which is predictably a direct result of valley hypsometry since glacier lengths in steeper valleys generally adjust less to equivalent changes in ELA. On the other hand, glaciers occupying the lower-gradient Lake and Clear creek valleys experienced generally higher reconstructed rates of retreat. Regardless, we find that while Pine Creek may have initiated ~500 yr sooner than the other two, all three valleys were in a period of ~1-1.5-kyr-long synchronous retreat once the other two glaciers began retreating. We conclude that while there may have been some hypsometric influences on the timing of deglaciation across our study site, evidence suggests these influences did not keep these glaciers from synchronously retreating during a majority of their deglaciation."

**Lines 285-288: Glaciers don't respond to $CO_2$, so directly comparing to $CO_2$ seems a little irrelevant. Of course, there is a close relationship between $CO_2$ and temperature, but why not compare your glacier retreat records to proxy global temperature (e.g. Shakun et al., 2012)?**

We thank the reviewer for highlighting this. We have changed the text and Figure 5 to show the proxy global temperature curve compiled in Shakun et al., 2012. We also recognize that the compilation curves from Shakun et al. are clearly influenced by more than just greenhouse gas forcing, particularly in the Northern Hemisphere. It may be argued that the southern hemisphere compiled record is more closely tied to atmospheric $CO_2$ concentrations so we are including both the global record and the southern hemisphere records in figure 5.

***Lines 327-334: The argument that there is similarity between records, and "possible teleconnections", isn't particularly convincing. The majority of the recorded retreat occurred before the North Atlantic climate shift; your ages indicate retreat initiated 1-2 kyr earlier that the climate shift at 14.7 ka. I'd like to see the text rephrased, without mention of teleconnections.***

As you observe, the glaciers in our field site do begin retreating prior to the onset of North Atlantic abrupt warming. Originally we were more focused on the fact that the rate and short-lived nature of retreat was most similar to the abrupt North Atlantic warming even though the timing was not perfect. And so, we removed mention of teleconnection since it is difficult to argue that glaciers retreated in response to N. Atlantic warming if they were already retreating prior to the abrupt warming event.

Rather, we reworded the section to read:

"We find that deglaciation at some locations in the southern Rocky Mountains encompasses the HS-1/Bølling transition. Furthermore, the relatively rapid and short-lived nature of retreat for glaciers in the Sawatch Range – and some others across the Southern Rocky Mountains – appears to be more consistent with the abrupt manner of warming observed in the North Atlantic. However, glaciers apparently were already retreating prior to the abrupt HS-1/Bølling transition at ~14.7 ka. Therefore, it is difficult to argue that North Atlantic warming alone forced glacier retreat in the Southern Rocky Mountains."

We also appended a few sentences in the abstract to read:

"Deglaciation in the Sawatch Range commenced ~2 – 3 kyr later than the onset of rising global $CO_2$, and prior to rising temperatures observed in the North Atlantic region at the Heinrich Stadial 1/Bølling transition."

***Lines 329-330: What is this period of relative glacier stability based on?***

Our original line of thought was that, based off of the previous moraine chronologies at our field site, it is possible that glaciers remained at relatively stable positions from the culmination of the LGM up until they began retreating at 16 – 15 ka. However, it is also possible that glaciers retreated in this time frame and then re-advanced to form the moraines deposited at 16 – 15 ka. We do not know which scenario is correct so we elected to remove this sentence.

***Line 383: "one of two", or both mechanisms, as you state below. Reword this, as these are not mutually exclusive explanations.***

We rephrased the sentence with the following: "we hypothesize that one of two – or a combination of both – possible regional climatic mechanisms…"

***Lines 385-386: As mentioned above, it is a difficult to accept that the glaciers "began retreating around the time of abrupt warming" when the data indicate retreat started at least 1-2 kyr before the climate transition. There is only correlation here if you doubt the accuracy (or precision) of the retreat ages, in which case you should discuss more thoroughly.***

Re-worded the text here to read as follows:

"First, we find that for some alpine glaciers in the region, the relatively rapid, short-lived and synchronous nature of retreat – including those in the Sawatch Range – across the southern Rocky Mountains is more consistent with the abrupt manner of warming observed in the North Atlantic than with increasing global temperature forced by $CO_2$ rise. However, evidence suggests glaciers were already retreating prior to the HS-1/Bølling transition."

***Lines 387-390: I like the comparison of the rates of glacier and climate change, as it makes use of your estimated retreat rates and it can be effective if there is any doubt in the absolute timing. However, a number of non-climatic, glaciological processes can also contribute to faster rates, even with a gradual forcing. Such processes also need to be considered.***

We agree that the rate of retreat can be modified by non-climatic factors. And this is in fact supported by the relationship between retreat rates and valley gradients that we previously discussed. However, that all three neighboring glaciers evacuated their valleys in the same 1-2 kyr interval in time, relatively quickly despite the variation in retreat rate, we believe must be climatically forced.

***Line 400: Sorry to be pedantic, but as above, glaciers don't respond to gas concentrations. Refer to global temperature instead.***

Changed the wording here and elsewhere to say, "increase in global temperature forced by $CO_2$ rise"

***Table 1: Transact distances are reported to the nearest metre over many thousands of metres. This seems unrealistically precise.***

As previously stated, we now round.

***Figure 2: Need to make clear what are the new data and what are previously published data. There are also two references to "n=x", which I presume need values added.***

Closed circles are now previously published ages and open circles are new ages in the figure.

The references to n = x in both cases have been resolved.

---

## Author Comment (AC3) · 22 Jul 2020

Reply to 'Reviewer comments' from RC 2 Benjamin Laabs

We thank RC 2 for their thoughtful and constructive review of our manuscript. We have gone through and replied to each individual comment. Below, please find the original comment bolded and italicized and our reply in normal font.

***Line-by-line comments:***

***Line 50: the Uinta Mountains are part of the Middle Rocky Mountains physiographic province and probably do not need to be singled out here (although they are awesome and have a fantastic glacial record).***

Removed the Uinta Mountains as a singled out entity

***Line 53: could probably state "Latest Pleistocene or Early Holocene" here, as Marcott et al. found that some cirque floor moraines were abandoned as early as 15 ka. Additionally, basal $^{14}$C ages from lake sediments inboard of cirque-floor moraines are Pleistocene in age in some mountains (see records published by J. Munroe for the Uinta Mountains (Munroe and Laabs, 2017) and by J. Munroe and others in the Ruby Mountains).***

Replaced "by the start of the Holocene" with "during the late glacial-to-early Holocene"

***Lines 90-93: should cite some of the earlier, original reports on the glacial record in southern Colorado and northern New Mexico. Jim McCalpin did some work in the region (mostly the Sangres) in the 1980s and Keith Brugger has done more recent mapping in the Sawatch.***

Thank you for bringing these citations to our attention. We felt it would be most appropriate to add the following citation to the list since we are only citing summary papers here:

Laabs, B. J. C., Licciardi, J. M., Leonard, E. M., Munroe, J. S., and Marchetti, D. W.: Updated cosmogenic chronologies of Pleistocene mountain glaciation in the western United States and associated paleoclimate inferences, Quaternary Science Reviews, 242, 106427, https://doi.org/10.1016/j.quascirev.2020.106427, 2020.

Although we did include the following citation to the section specifically discussing the Sawatch Range:

Brugger, K. A., Ruleman, C. A., Caffee, M. W., and Mason, C. C.: Climate during the Last Glacial Maximum in the Northern Sawatch Range, Colorado, USA, Quaternary, 2, 36, 2019a.

***Lines 101-104: the Guido et al. cosmo ages are pre-CRONUS (and also pre-really***

*good AMS measurements) and probably should be recalculated in order to accurately compare with more recently published cosmo ages from southern Colorado. If you've already done this, then it's worth specifying here. If not, the Guido et al. ages are available in ICE-D.*

Thank you for the suggestion. we recalculated the ages and updated some of the text to reflect those changes. In addition, we added the following phrase at the beginning of the section:

"ages discussed below are re-calculated using the promontory point production rate calibration of Lifton et al. (2015) and the LSD$n$ scaling model of Lifton et al. (2014)"

*Line 174: prior to this paragraph, consider adding a paragraph about how exposure ages of glacially scoured bedrock are related to ice margin position and some potential limitations of dating these to track ice retreat compared to moraines. As you know, glacially scoured bedrock surfaces that protrude above the valley floor (forming smooth and easy-to-sample surfaces) represent places of minimal scour depth, which can result in an inheritance problem. The Bayesian approach helps to sort this out by accounting for relative age differences, but even so, the potential for inheritance is greater than for most other applications of cosmogenic dating and should be acknowledged. Snow cover is another important consideration along valley floors and should be acknowledged if not assessed.*

Thank you for the suggestion. We added the following sentence to the end of the first paragraph in the section to acknowledge that incomplete erosion is an issue when it comes to exposure ages on glacially sculpted bedrock:

"Bedrock surfaces located in the bottoms of valley floors – where glacial erosion is maximized – were specifically targeted since the potential for incomplete scouring of these surfaces can lead to inherited nuclides and ages that are older than expected."

In addition, we did not choose to make any corrections for snow shielding nor post-depositional bedrock erosion and we acknowledge that with the following sentence at the end of the second paragraph in the section:

"We do not attempt to make any corrections for snow cover or post-depositional bedrock surface erosion."

*Lines 189-211: consider reorganizing the reporting of ages here. The bedrock exposure ages are reported first, then the exposure ages of recessional moraines/young modes of terminal moraines, and then the bedrock ages are described again. Perhaps starting with the moraine ages (or including them in a previous section) and focusing just on the bedrock exposure ages here would improve the flow of this section and a smoother transition to the retreat rates in the subsequent paragraph.*

Thank you for the suggestion. We agree that it is a little awkward reporting the moraine ages here in the results when we did not date them in this study. So we decided to move this paragraph to the previous section (2. Setting).

***Lines 233-242: the statistical reasons for excluding four exposure ages are explained well here, but the most likely reason that some exposure ages fall out of stratigraphic order, inconsistent exposure between sample sites, is not. As noted in a previous comment, the potential limitations of bedrock exposure ages should be acknowledged.***

We added in the following sentences to the end of the paragraph that hopefully convey the potential issues associated with each suspected outlier:

"Two suspected outliers are older than expected, which may have been caused by insufficient glacial erosion leading to inheritance. The two remaining potential outliers are younger than expected, which could have resulted from excessive soil and snow cover, enhanced post-depositional bedrock surface erosion, or erosion and removal of overlying sediments, or a combination of these factors."

***Lines 243-255: I can't see the reason for using NENA-Lm as an example of another production/scaling model for high altitude sites in western NA. The NENA calibration site is far away and much lower in elevation, and I think the reason for using it in some earlier studies in the mountain west was to illustrate the effects of lower SLHL production rate (which started to appear in the literature circa 2010) on exposure ages. Perhaps a better option would be to compare the ages computed with the Promontory Point calibration/LSDn scaling with ages computed with a globally averaged production rate and LSDn scaling, or just show the effects of using different scaling models with the Promontory Point calibration? This would better illustrate the degree to which the choice of production rate affects exposure ages, which I assume is what the authors are doing here.***

The goal of using NENA here was to show it as a sort of an 'end-member' production rate since it is relatively low (especially compared to PPT). In addition, it is the other production rate (or series of rates) that exist from North America. We did originally calculate the ages using the default PR in CRONUS and those ages fall somewhere between PPT and NENA. We wanted to emphasize that even if we used a relatively low PR that produces ages which are 9 – 12% older (e.g. NENA), there is still a significant delay in deglaciation compared to the time of global warming and $CO_2$ rise.

We rephrased the beginning of the final paragraph in section 5.1 to read:

"Although we interpret our results using the Promontory Point production rate calibration site (Lifton et al., 2015) and the LSD$n$ scaling scheme (Lifton et al. 2014), we calculate exposure ages with other commonly used calibration sites for North America (e.g.

northeastern North America NENA; Balco et al., 2009 and the 'global' production rate; Borchers et al., 2016) and another commonly used scaling scheme (Lal/Stone–Lm; Lal, 1991; Stone, 2000). Samples used for the NENA production rate calibration range in elevation between ~50 to 400 m asl and are located ~3000 km northeast of the Sawatch Range. This combination produces the oldest ages given the previously mentioned reasonable production rate calibrations and scaling schemes, and are between 9 to 12% older than when using PPT/LSDn (all other combinations fall somewhere in between; Fig. 4; Table 1)."

***Lines 267-269: should probably cite Young et al. (2011) at the end of this sentence.***

Done

***Fig. 1: this is a beautiful map! As you reference some other glaciated mountains in the western U.S. in the introductory paragraphs, consider labeling some of the ones shown on the map along with pluvial lakes.***

We originally chose not to label all of the other glacial centers in the western US since we did not discuss chronologies from any other location outside the southern Rocky Mountains.

***Fig. 2: seems like a good idea to show all the terminal moraine cosmo ages instead of just the young mode at Pine Creek, given that the terminals are the "starting point" for ice retreat? Just a suggestion; I understand that you're emphasizing the onset of ice recession in this paper, not the glacier maxima.***

While we do agree that this might be valuable, we did not originally report individual moraine ages in the text (rather an approximate age range) for the culmination of the LGM in our field area since, as you mention here, we are only focusing on deglaciation rather than the LGM. That said, we will report the LGM moraine ages on the figure and mention in caption that the ages are mean ages from moraine boulders reported in Schweinsberg et al. (2020).

Lake Creek terminal moraine: 20.6 ± 0.6 ka
Clear Creek terminal moraine: 20.0 ± 1.0 ka
Pine Creek terminal moraine: 22.3 ± 1.3 ka

***Fig. 5: may want to consider a more recent and focused assessment of the Bonneville hydrograph in Oviatt (2015) or some of the specific discussions about the duration of the Provo phase of the lake by D. Miller (2016).***

We did not find large enough differences between the Reheis et al. (2014) lake level curves and those from Oviatt et al. (2015) and D. Miller (2016) that would significantly alter our interpretations since the timing of North American ice sheet separation, lake level lowering and Sawatch Range deglaciation are currently only loosely correlated.

---

## Author Comment (AC4) · 22 Jul 2020

Reply to 'Comments by AE' from EC 1 Yeong Bae Seong

We thank the associate editor for their consideration of our manuscript and their constructive comments. Below, please find the original comments bolded and italicized and our reply in normal font.

***Ln 271-2: "We find that all three valley glaciers did not begin significantly retreating until ~5 – 6 kyr after the culmination of the LGM in the Sawatch Range". You cannot jump over the gun like above because you are based on the different types of samples (i.e. Moraine erratics VS bedrock). Moraine boulders can indicate advance or stagnation (As you know there is some debate on the implication of ages of boulders on a moraine. Polished bedrock usually indicates the timing of deglaciation as you did. There should be differentiation on the interpretation of ages of two types sampled (Reviewer 2 told about this problem). You may want to make more explanation and discussion on this matter.***

We, along with other groups working on moraine dating, interpret moraine boulder ages as the culmination of a glacier advance. We think that the boulders on top of a moraine are the last to be deposited, thus represent the end of the advance. And when the moraine, and the uppermost layer of sediment (the surface boulders that we often choose to date), becomes abandoned, the boulder clocks begin. Thus, we think that the 16 ka moraines in our study valleys ought to be a suitable starting point for our up-valley bedrock transects.

We added in the following phrase to the highlighted sentence:

"We find that all three valley glaciers did not begin significantly retreating until ~5 – 6 kyr after the culmination of the LGM in the Sawatch Range (since we assume boulder ages on LGM moraines represent the timing of moraine abandonment)."

***Figure 2: Can you separate the type of samples for $^{10}$Be dating? For example, erratics (open circle) Vs bedrock (filled circle).***

In figure 2, all of the samples with circles are from sculpted bedrock. The only ages from moraine boulders are the ones that are averages for the recessional moraines in Lake Creek and younger mode of ages on the terminal moraine in Pine Creek. We thank you for the comment, it will help clarify for our readers should the manuscript be accepted.

We changed the label for the moraine ages in all three valleys to better distinguish moraine boulder ages from sculpted bedrock ages.

***Figure 3: How about showing the sample number on the picture (or on the sampled boulder or bedrock), which is better to readers?***

Great – this also helps to clarify. We now include sample names in the images along with the reported ages.